# Role of a Chest X-ray Severity Score in a Multivariable Predictive Model for Mortality in Patients with COVID-19: A Single-Center, Retrospective Study

**DOI:** 10.3390/jcm11082157

**Published:** 2022-04-12

**Authors:** Masoud Baikpour, Alex Carlos, Ryan Morasse, Hannah Gissel, Victor Perez-Gutierrez, Jessica Nino, Jose Amaya-Suarez, Fatimatu Ali, Talya Toledano, Joseph Arampulikan, Menachem Gold, Usha Venugopal, Anjana Pillai, Kennedy Omonuwa, Vidya Menon

**Affiliations:** 1Department of Radiology, Harvard Medical School, Massachusetts General Hospital, 55 Fruit Street, Boston, MA 02114, USA; mbaikpour@mgh.harvard.edu; 2Department of Medicine, NYC Health and Hospitals/Lincoln, 234 East 149th Street, Bronx, NY 10451, USA; carlosa@nychhc.org (A.C.); morasser@nychhc.org (R.M.); perezgv@nychhc.org (V.P.-G.); ninomoj@nychhc.org (J.N.); amayasj@nychhc.org (J.A.-S.); fatimatu.ali@nychhc.org (F.A.); usha.venugopal@nychhc.org (U.V.); anjana.pillai@nychhc.org (A.P.); kennedy.omonuwa@nychhc.org (K.O.); 3Department of Interventional Radiology, George Washington University Hospital, 900 23rd Street NW, Washington, DC 20037, USA; hannah.gissel@gmail.com; 4Department of Radiology, NYC Health and Hospitals/Lincoln, 234 East 149th Street, Bronx, NY 10451, USA; talya.toledano@nychhc.org (T.T.); joseph.arampulikan@nychhc.org (J.A.); menachem.gold@nychhc.org (M.G.)

**Keywords:** COVID-19, chest X-ray, severity score, predictive model, mortality

## Abstract

Predicting the mortality risk of patients with Coronavirus Disease 2019 (COVID-19) can be valuable in allocating limited medical resources in the setting of outbreaks. This study assessed the role of a chest X-ray (CXR) scoring system in a multivariable model in predicting the mortality of COVID-19 patients by performing a single-center, retrospective, observational study including consecutive patients admitted with a confirmed diagnosis of COVID-19 and an initial CXR. The CXR severity score was calculated by three radiologists with 12 to 15 years of experience in thoracic imaging, based on the extent of lung involvement and density of lung opacities. Logistic regression analysis was used to identify independent predictive factors for mortality to create a predictive model. A validation dataset was used to calculate its predictive value as the AUROC. A total of 628 patients (58.1% male) were included in this study. Age (*p* < 0.001), sepsis (*p* < 0.001), S/F ratio (*p* < 0.001), need for mechanical ventilation (*p* < 0.001), and the CXR severity score (*p* = 0.005) were found to be independent predictive factors for mortality. We used these variables to develop a predictive model with an AUROC of 0.926 (0.891, 0.962), which was significantly higher than that of the WHO COVID severity classification, 0.853 (0.798, 0.909) (one-tailed *p*-value = 0.028), showing that our model can accurately predict mortality of hospitalized COVID-19 patients.

## 1. Introduction

The unprecedented spread of Coronavirus Disease 2019 (COVID-19) infection across the world has strained our society with significant morbidity and mortality. Its clinical manifestations range from minor flu-like symptoms to severe pneumonia that can lead to acute respiratory distress syndrome, multiple organ dysfunction syndrome, and death [1]. The role of imaging has evolved through the pandemic, with radiologic studies playing an important role in the diagnosis and determination of disease severity [2]. Even though Computerized Tomography (CT) scans have the highest sensitivity for the characterization of pulmonary involvement in COVID-19 disease [3,4,5], factors such as easy accessibility, higher logistical costs, time management, radiation considerations, and compliance with infection control measures are barriers to its widespread use as the primary imaging modality in the setting of a pandemic [2,6,7]. As an alternative, a chest X-ray (CXR) has been shown to play an essential role in the initial evaluation and risk stratification of patients, with some variability in its reported sensitivity, which is influenced by the timing of the onset of symptoms [7,8]. Compared to a CT scan, this imaging modality is widely available, is cheaper, exposes the patient to less radiation, and can be obtained in a timely manner at bedside with quicker and easier disinfection procedures [9,10,11].

One of the main problems with using imaging modalities as diagnostic tools for COVID-19 is that all radiological findings in this disease are generic and have significant overlap with other infectious processes. Findings such as reticulonodular or ground-glass opacities, patchy infiltration, and consolidation can be identified on CT scans as early as the fifth day following onset of symptoms, while changes in CXRs are expected to appear on days 10 to 12 [2,12]. CXRs of patients with moderate COVID-19 pneumonia demonstrate patchy opacities and reticulonodular infiltration. In severe or critical infections, patients develop patchy alveolar infiltrations that confluence over time, forming consolidations that correlate with acute respiratory distress syndrome [13]. Multiple retrospective studies have evaluated the association between an initial CXR and clinical outcomes in patients with COVID-19 pneumonia. Orsi et al., presented a scoring system based on the extent of lung involvement on CXRs ranging from 0 to 8 and, although they did not evaluate the correlation of their scoring system with patient outcome, they showed a significant positive correlation with C-reactive protein, lactate dehydrogenase, and fever duration, and a negative correlation with oxygen saturation [14]. Toussie et al., developed a CXR scoring system based on the opacities in six separate lung zones, ranging from 0 to 6, with scores >2 associated with hospitalization and >3 associated with intubation [15]. Using a similar scoring system, Borghesi et al., showed that CXR findings correlated with mortality in a large Italian cohort [16]. The radiology assessment of lung edema (RALE) is another scoring system ranging from 0–48, based on CXR findings. In their study on COVID-19 patients, Cozzi et al., showed that with each unit increase in the RALE score, the hazard for death increased by 1.23 [17].

Considering our unique patient population in the South Bronx is predominantly composed of minorities with the highest prevalence of chronic diseases such as hypertension, diabetes, obesity, and COPD among the five boroughs of New York City [18], we aimed to develop a simplified scoring system based on baseline CXR findings and assess its predictive value for the mortality of hospitalized COVID-19 patients.

## 2. Materials and Methods

### 2.1. Study Design and Sample Population

This single-center, retrospective, observational study was reviewed and approved by the institutional review board (IRB# 20-007), and the requirement for obtaining written consent from the participants was waived. Consecutive adult patients admitted to the hospital from 5 March to 16 April 2020 with a positive RT-PCR test for SARS-CoV-2 on nasopharyngeal swab samples (Bio-Reference Laboratories, Inc., Elmwood Park, NJ, USA) and a chest X-ray performed in the emergency room were included in the study. Radiological findings better explained by non-COVID-19 causes such as acute congestive lung markings, effusion, abscess, white-out lung, or pneumothorax were excluded from our study. Subjects with documented pulmonary diseases such as interstitial lung disease, sarcoidosis, tuberculosis, lung cancer or metastasis to lung, or prior pneumonia within the last 3 months were also excluded (Figure 1).

### 2.2. Data Collection

Baseline characteristics, comorbidities, clinical data on admission, and clinical course were collected from electronic medical records for each patient. Collected data included age, gender, race, body mass index, smoking, medical comorbidities, symptoms on admission (fever, cough, shortness of breath, gastrointestinal symptoms, and altered mental status/seizures), and the number of days from the onset of symptoms to admission. Oxygen requirement data were taken from the first encounter in the emergency department, including pulse oximetric saturation (SpO_2_), fraction of inspired oxygen (FiO_2_), the SpO_2_/FiO_2_ ratio (S/F ratio), and the need for mechanical ventilation. Sepsis syndrome on admission was assessed based on the quick Sequential Organ Failure Assessment (qSOFA) score [19]. COVID-19 severity based on the Chinese CDC criteria [10], intubation during hospital course, intubation duration, length of stay, and outcome at discharge were also recorded for each patient.

### 2.3. Imaging and Analysis

Chest radiographs acquired in the anteroposterior (AP) view using a portable X-ray machine during their emergency department stay were examined retrospectively. These were read by three radiologists with 12, 15, and 15 years of experience in thoracic imaging, who were blinded to the patients’ clinical history and condition. They were asked to give a consensus reading of two radiologic features: distribution and characteristics of the lung opacities. They assigned a score from 0 to 4 for the distribution of lung opacities: 0 for normal; 1 for unilateral–unilobar; 2 for unilateral–multilobar; 3 for bilateral–not diffuse; and 4 for diffuse bilateral (Figure 2, Figure 3, Figure 4, Figure 5, Figure 6 and Figure 7). They also scored the characteristics of the lung opacities from 0 to 2: 0 for normal; 1 for hazy or interstitial; 2 for dense opacities or any opacities with dense component (hazy–dense, interstitial–dense). A CXR score was then derived from the product of these two radiologic features, with the final score ranging from 0 to 8.

### 2.4. Statistical Analysis

Categorical variables were summarized as frequency (percentage) and continuous variables were presented as either mean (standard deviation—SD) or median (interquartile range—IQR) according to their distribution.

Binary logistic regression analysis was used to assess the correlation between possible confounders and outcome at discharge to identify independent predictive factors for in-hospital mortality in these patients. The results are presented as odds ratios (ORs) and 95% confidence intervals. The dataset was then randomly split into derivation and validation datasets with a ratio of 70:30, respectively. Binary logistic regression analysis was used on the derivation dataset to develop a model with the identified factors to predict in-hospital mortality. The model was then tested on the validation dataset. Its predictive value was calculated as the area under the receiver operating characteristic curve (AUROC) and then compared to that of the COVID severity using Hanley and McNeil’s method [20], acknowledging that the COVID severity score was not specifically designed as a predictive factor for the mortality of COVID-19 patients.

All the analyses were performed using IBM SPSS statistical software version 25 (IBM SPSS Corp., Armonk, NY, USA). A *p*-value less than 0.05 was considered statistically significant.

## 3. Results

A total of 628 patients admitted with a confirmed diagnosis of COVID-19 were included in this study. Patients’ baseline characteristics, underlying comorbidities, and clinical course are shown in Table 1. The majority of the patients were male (58.1%), Hispanic (65.1%), and obese (50.3%). The mean age was 60 ± 16 years old. The most prevalent comorbidity was hypertension (44.6%) followed by diabetes (42.8%). Regarding symptomatology, 71.2% reported a cough, 69.3% reported shortness of breath, and 65.0% reported a fever. The median S/F ratio was 303.3 (IQR, 102.1–447.6). Sepsis syndrome was present on admission in 30.1% of patients. In terms of the severity of COVID-19 pneumonia, 41.2% were classified as moderate, 28.3% as severe, and 30.4% as critical according to the Chinese CDC criteria as described in the Section 2. Some 134 patients (21.3%) required intubation on admission, and 117 patients (18.7%) required intubation during their hospital course.

Abnormal CXR findings on admission were observed in 70.6% (479) of our cohort. Common CXR findings included hazy or interstitial infiltrates in 74.5% (357), while dense consolidation was seen in 22.5% (122). In terms of distribution, diffuse bilateral infiltrates were predominant and seen in 69.1% (331), followed by unilateral–unilobar in 13.4% (64), and bilateral—not diffuse in 13.2% (63) of cases. The median CXR severity score for our cohort was 3 (IQR, 1–4). More than half of the patients (53.4%) had diffuse bilateral changes with hazy opacities, which corresponds to a CXR score of 4, followed by diffuse bilateral with dense opacities in 15.7% (75), which corresponds to a CXR score of 8. The distribution and frequency of radiographic findings are shown in Table 2.

Binary logistic regression analysis was used to identify independent predictors for in-hospital mortality in our cohort. The results of this analysis, presented in Table 3, found age (*p* < 0.001), sepsis on admission (*p* < 0.001), S/F ratio (*p* < 0.001), mechanical ventilation on admission (*p* < 0.001), and the CXR severity score (*p* = 0.005) to be independent predictive factors for the in-hospital mortality of patients.

The dataset was then randomly split into two derivation (*N* = 439) and validation (*N* = 189) datasets with a ratio of 70:30, respectively. According to the binary logistic regression analysis results, a model was developed on the derivation dataset and included the following variables: age, sepsis syndrome on admission, S/F ratio, mechanical ventilation on admission, and CXR severity score. The specifics of the model are presented in Table 4. The model was then tested on the validation dataset, yielding an AUROC of 0.926 (95% CI: 0.891, 0.962). The AUROC of COVID severity was also calculated using the same dataset (*N* = 189) as 0.853 (95% CI: 0.798, 0.909). Hanley and McNeil’s method was used in a one-tailed setting, testing to see if the AUROC of our predictive model was more significant than that of the WHO COVID severity. The results showed our model to have a significantly higher predictive value compared to COVID severity, with a one-tailed *p*-value of 0.028 (Figure 8). Although COVID severity score was not specifically designed as a predictive factor for the mortality of COVID-19 patients, it showed a relatively high predictive value for in-hospital mortality in our patient population, even compared to the AUC values reported in the literature for models designed specifically for predicting mortality.

## 4. Discussion

Considering the limited medical resources available during the COVID-19 pandemic surge, the accurate prediction of patients with COVID-19 who are particularly at risk for poor outcomes can be very valuable in determining how to best allocate resources and optimize care. Accordingly, numerous studies have used clinical data, lab results, and information obtained from imaging to develop statistical and machine-learning models to predict the mortality of COVID-19 patients [5,6,7,8]. In the present study, we defined a simple CXR severity scoring system based on the extent of lung involvement and density of lung opacities and combined it with four readily available clinical variables on admission (age, qSOFA score ≥ 2, S/F ratio, and need for mechanical ventilation on admission) to develop a linear regression model in our derivation cohort of 439 patients. We then tested the model in our validation dataset (*N* = 189), yielding an AUROC of 0.926 (95% CI: 0.891, 0.962) for predicting mortality in hospitalized COVID-19 patients.

Although prior studies have introduced comparable CXR scoring systems, to our knowledge, this study is one of the first to have incorporated a CXR severity scoring system in a multivariable model to provide an accurate predictive tool that can be used at bedside on admission. For instance, Toussie et al., designed a scoring method in which each lung was divided into three zones and each zone was given a score of 0 if opacity was absent or 1 if opacity was present. The sum of these scores constituted the final score, with scores greater than 2 associated with hospitalization and those greater than 3 associated with intubation. In their cohort of 338 patients, only 7% expired throughout the study and—although most of these patients had more extensive lung opacification—a meaningful relationship could not be demonstrated [15]. In another study, Borghesi et al., introduced their “Brixia” CXR scoring system in which they similarly divided the lungs into six zones on frontal chest projection, but instead of giving a binary score to each zone, they gave a score of 0 to 3, with 0 representing no lung abnormalities, 1 for interstitial infiltrates, 2 for interstitial and alveolar infiltrates with interstitial predominance, and 3 for alveolar predominance. Their univariable analysis in a sample of 100 hospitalized patients showed a significantly higher CXR score in patients who expired compared to those who were discharged [16]. In a subsequent study, they incorporated their Brixia score in a multivariable regression model on a sample population of 302 patients and found the Brixia score, age, and conditions that induced immunosuppression to be independent predictive factors of in-hospital mortality. Their study had considerable statistical limitations as they used the same population to identify relevant covariates, develop their model, and test its predictive value [21], as opposed to the current study, in which we divided our sample population into two subsets for derivation and testing. They also used the chest radiograph of the patients with the highest Brixia score, which limits the application of their model at presentation of the patients to the hospital. Garrafa et al., used the Brixia score in developing and testing a machine-learning model based on the random forests classification algorithm for predicting in-hospital mortality. Based on their results, their model of age, Brixia score, and multiple blood analytes had an AUC of 0.78 for predicting in-hospital mortality in their testing dataset of 676 patients [22]. Although they used a larger sample population for both developing and testing their model, and included patients from both the first and second waves of the COVID-19 pandemic, the predictive performance of their model was much lower than our logistic regression model with an AUC of 0.93, which may be attributed to the fact that they did not use important clinical variables such as oxygen requirements, sepsis syndrome, or the need for mechanical ventilation. The Radiographic Assessment of Lung Edema (RALE) score, initially proposed by Warren et al., in 2018 [23], was another tool that was adapted, modified, and used for the prediction of different outcomes in COVID-19 patients. In their study on 90 COVID-19 patients, Homayounieh et al., reported a significantly higher RALE score in deceased patients (*N* = 21) compared to the surviving subjects (*N* = 69) [24]. Joseph et al., used a modified RALE scoring system (scores assigned to each lung instead of four quadrants) and reported higher scores being associated with a higher risk of composite adverse clinical outcome of intubation, ICU admission, or death [25]. Neither of these studies provided a measure of accuracy for their CXR scoring systems in predicting the mortality of patients, nor did they include their scoring methods in a multivariable model to develop a predictive model.

Later, Balbi et al., used the Brixia score in their study conducted on a sample of 340 COVID-19 patients. They ran a binary logistic regression analysis and found that the Brixia score, age, PaO2/FiO2 ratio, and cardiovascular diseases were independent predictive factors for death in these patients [26]. They did not present the results of this analysis as a combined predictive model, so no direct comparison can be made between our findings and theirs, but the independent variables in both studies were comparable. In a more comprehensive study performed on a sample of 356 hospitalized COVID-19 patients, Schalekamp et al., proposed another method in which the chest radiograph was divided into four zones and the extent of lung involvement was scored as 0 for no involvement; 1 for estimated involvement of lung parenchyma 0–50%; and 2 for estimated involvement >50%. The final score ranged between 0 and 8. They further combined this CXR score in a multivariable model including variables of sex, chronic obstructive lung disease, symptom duration, neutrophil count, C-reactive protein level, lactate dehydrogenase level, and distribution of lung disease (diffuse vs. peripheral) and introduced the “Dutch COVID-19 risk model” for predicting critical illness. Their model yielded an AUROC of 0.77 (95% CI: 0.72, 0.81; *p* < 0.001) [27]. In comparison, our model has a higher predictive performance for in-hospital mortality rather than critical illness, includes readily available clinical variables rather than lab values, was developed using a larger sample population, and was tested on a validation dataset different from the derivation cohort. On the other hand, our study was conducted in a single center, while the data for the Dutch COVID-19 risk model came from two Dutch community hospitals. It should also be kept in mind that there was a notable difference between the two sample populations with a 36.0% death rate in our cohort compared to 27.2% in their population. This could be attributed to the differences in the baseline comorbidities of the patients.

To summarize the strengths of our work compared to other similar studies, we had a larger sample population, we developed an accurate predictive model for in-hospital mortality in COVID-19 patients and validated it in a smaller sample of patients admitted to our center, we used easily accessible clinical variables along with information obtained from CXRs making it suitable for bedside use, and our model had a considerably high predictive value for the outcome, which was superior or comparable to other statistical and machine-learning models [28,29,30,31,32].

There were also several limitations to this work. Our prediction model was only internally validated, and it is based on a retrospective cohort of patients from one hospital providing healthcare to a particular population with a high prevalence of baseline comorbidities contributing to a higher death rate due to COVID-19, which limits its generalizability. Additionally, the data for this study were collected during the first wave of the COVID-19 pandemic, and since then, multiple new variants have emerged, which raises concerns about the performance of our model on new COVID-19 patients afflicted with the new variants. However, the variables we included in our model are conceptually general indicators of poor outcome in any acute respiratory disease, or even non-respiratory acute conditions. Another limitation was that all clinical parameters and laboratory values known to be associated with worse outcomes were not included in our model, mainly because they were not available for all patients. Moreover, the three radiologists reading the CXRs reached a consensus and provided us with a final score, which does not reflect the actual clinical workflow, and consequently, inter-rater variability could not be assessed in the setting of our study.

In summary, we found that basic clinical information and a simple assessment of lung involvement on a CXR—which are available in the first few hours of hospital admission—can provide complementary information for the prognosis of hospitalized patients with COVID-19. We demonstrated that a simple model composed of age, sepsis syndrome on admission, S/F ratio, mechanical ventilation on admission, and a CXR severity score were predictive of in-hospital mortality among these patients.

## Figures and Tables

**Figure 1 jcm-11-02157-f001:**
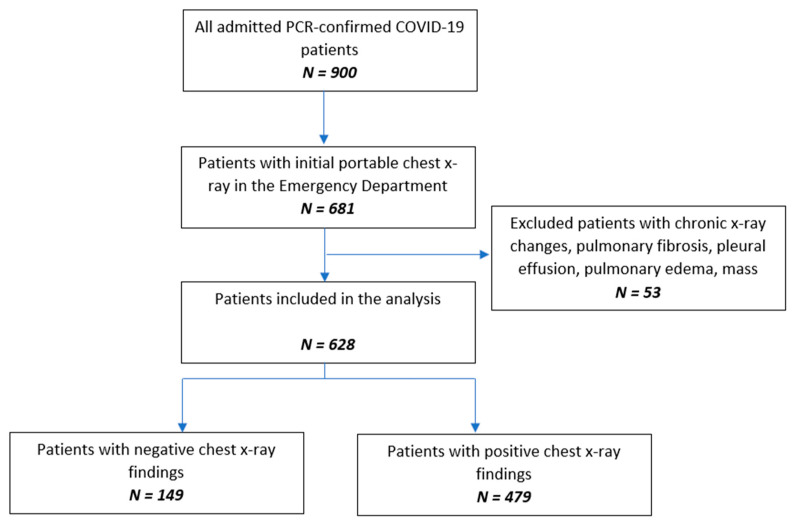
Flow chart of the study population.

**Figure 2 jcm-11-02157-f002:**
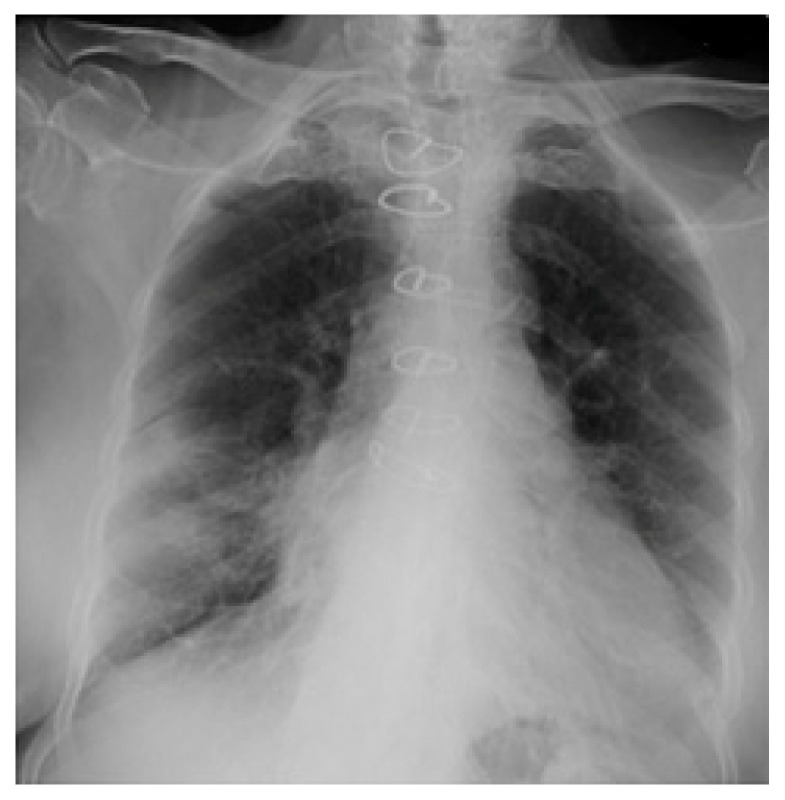
Unilateral–unilobar—hazy, right lower lobe infiltrate.

**Figure 3 jcm-11-02157-f003:**
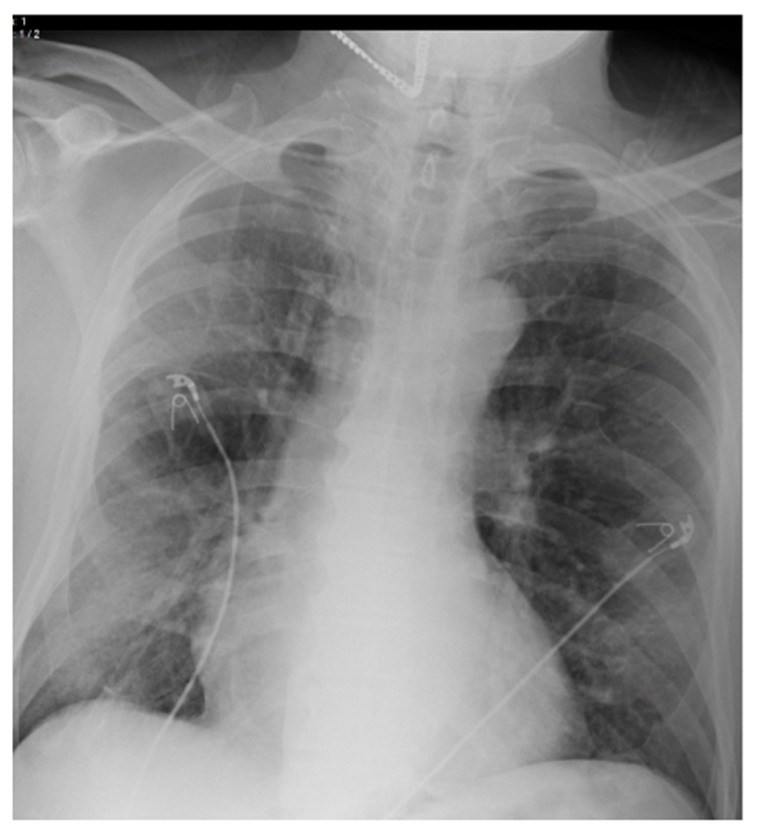
Unilateral–multilobar—hazy infiltrates throughout the right lung.

**Figure 4 jcm-11-02157-f004:**
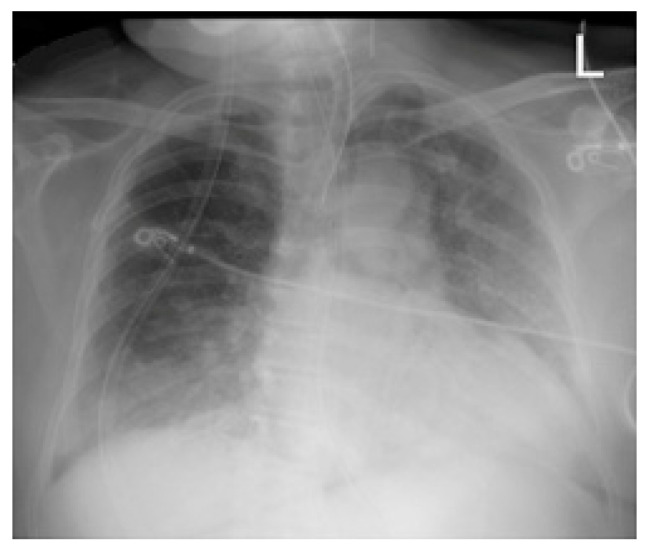
Bilateral—not diffuse—hazy infiltrates with a left predominance.

**Figure 5 jcm-11-02157-f005:**
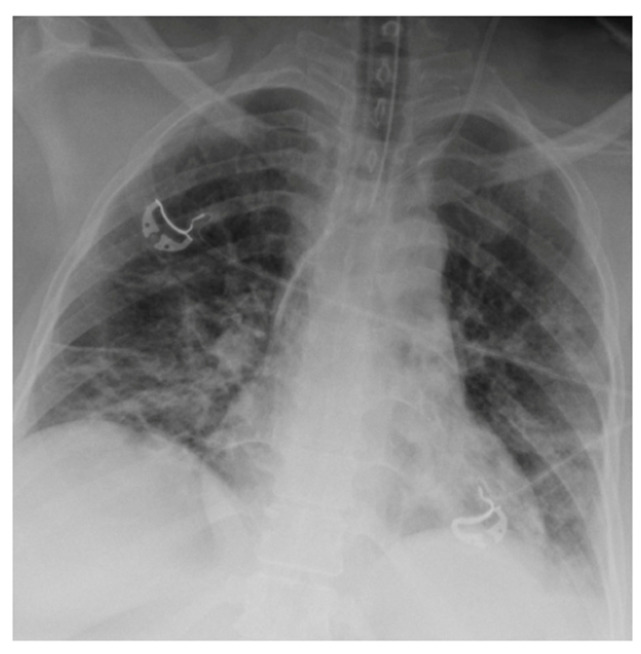
Diffuse bilateral—hazy infiltrates throughout both lungs.

**Figure 6 jcm-11-02157-f006:**
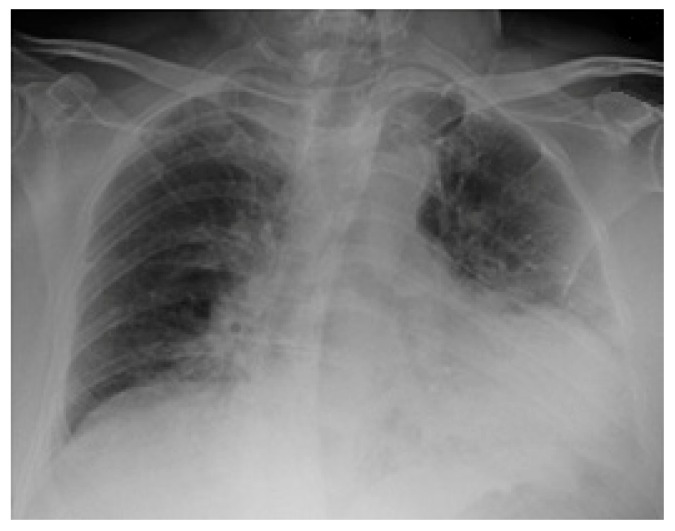
Dense infiltrate in the left lower lobe.

**Figure 7 jcm-11-02157-f007:**
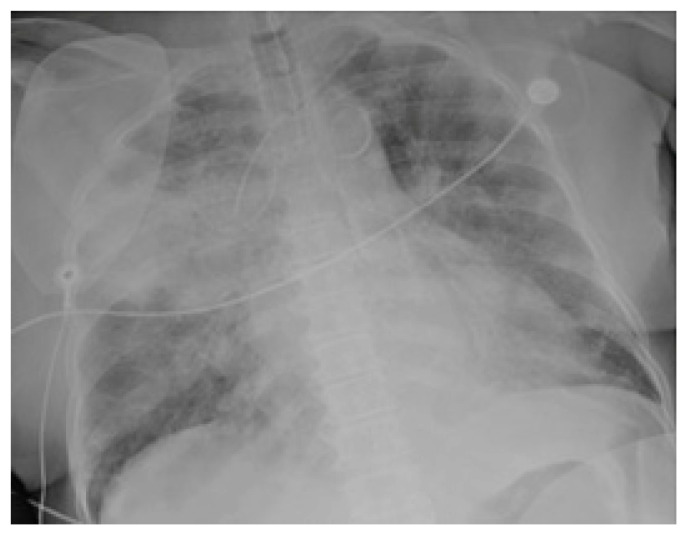
Dense infiltrate in the right upper lobe.

**Figure 8 jcm-11-02157-f008:**
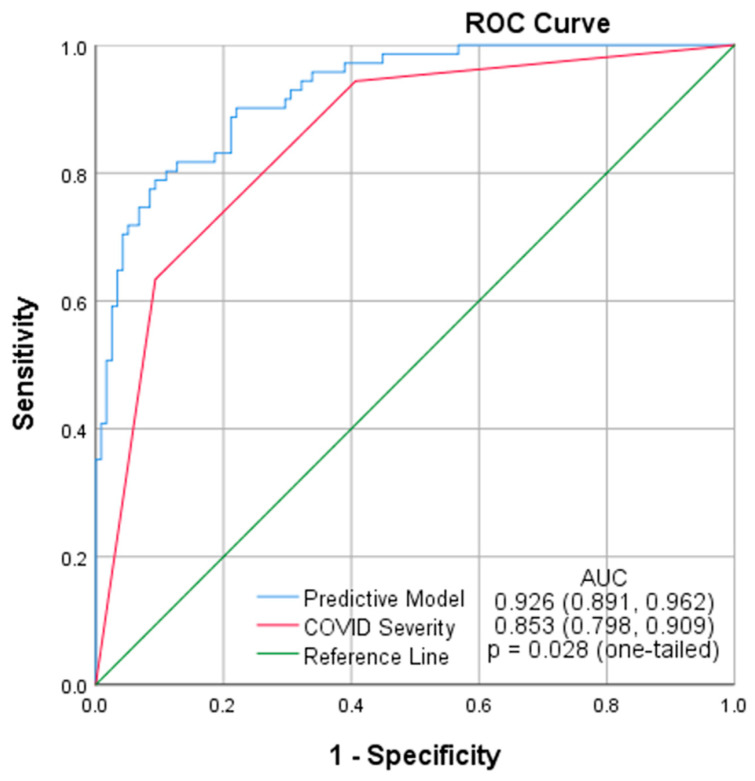
Comparing the AUROC of the predictive model with that of COVID severity in predicting mortality of COVID-19 patients in the validation dataset (*N* = 189).

**Table 1 jcm-11-02157-t001:** Descriptive statistics of the variables included in the study.

Variable		Total (*N* = 628)	Status at Discharge	
			Alive (*N* = 402)	Expired (*N* = 226)
Age *		59.6 (16.0)	55.4 (15.7)	67.1 (13.9)
Gender	Female	263 (41.9%)	174 (43.3%)	89 (39.4%)
	Male	365 (58.1%)	228 (56.7%)	137 (60.6%)
Race	Hispanic	409 (65.1%)	252 (62.7%)	157 (69.5%)
	African American	165 (26.3%)	112 (27.9%)	53 (23.5%)
	White	21 (3.3%)	16 (4.0%)	5 (2.2%)
	Asian	13 (2.1%)	9 (2.2%)	4 (1.8%)
	Other	20 (3.2%)	13 (3.2%)	7 (3.1%)
Smoking		37 (6.0%)	27 (6.8%)	10 (4.6%)
BMI	Underweight	7 (1.2%)	3 (0.8%)	4 (1.9%)
	Normal	95 (16.3%)	62 (16.7%)	33 (15.7%)
	Overweight	187 (32.1%)	126 (33.9%)	61 (29.0%)
	Obese	293 (50.3%)	181 (48.7%)	112 (53.3%)
Past Medical History				
Asthma/COPD		112 (17.8%)	67 (16.7%)	45 (19.9%)
Hypertension		280 (44.6)	182 (45.3%)	98 (43.4%)
CHF		47 (7.5%)	23 (5.7%)	24 (10.6%)
CKD		76 (12.1%)	40 (10.0%)	36 (15.9%)
DM		269 (42.8%)	152 (37.8%)	117 (51.8%)
Rheumatological Diseases		22 (3.5%)	10 (2.5%)	12 (5.3%)
Cirrhosis		7 (1.1%)	6 (1.5%)	1 (0.4%)
Transplant		4 (0.6%)	2 (0.5%)	2 (0.9%)
Immunosuppression		14 (2.2%)	8 (2.0%)	6 (2.7%)
HIV		16 (2.5%)	11 (2.7%)	5 (2.2%)
Cancer		39 (6.2%)	15 (3.7%)	24 (10.6%)
Symptoms				
Fever		408 (65.0%)	275 (68.4%)	133 (58.8%)
Cough		447 (71.2%)	302 (75.1%)	145 (64.2%)
Shortness of Breath		435 (69.3%)	254 (63.2%)	181 (80.1%)
Gastrointestinal Symptoms		141 (22.5%)	110 (27.4%)	31 (13.7%)
Altered Mental Status/Seizures		80 (12.7%)	32 (8.0%)	48 (21.2%)
Days from Onset of Symptoms **		4.0 (2.0, 7.0)	5.0 (3.0, 7.0)	4.0 (2.0, 7.0)
Sepsis Syndrome		189 (30.1%)	53 (13.2%)	136 (60.2%)
S/F ratio **		303.1 (102.1, 447.6)	342.9 (263.9, 457.1)	102.2 (97.0, 266.7)
COVID Severity	Moderate	259 (41.2%)	247 (61.4%)	12 (5.3%)
	Severe	178 (28.3%)	112 (27.9%)	66 (29.2%)
	Critical	191 (30.4%)	43 (10.7%)	148 (65.5%)
Length of Stay **		6.0 (3.0, 11.0)	5.0 (2.0, 10.0)	7.0 (4.0, 11.0)
Mechanical Ventilation on Admission		134 (21.3%)	28 (7.0%)	106 (46.9%)
Mechanical Ventilation		251 (40.0%)	67 (16.7%)	184 (81.4%)
Days to Intubation **		0.0 (0.0, 3.0)	3.0 (0.0, 8.0)	0.0 (0.0, 2.0)
Duration of Mechanical Ventilation **		7.0 (4.0, 12.0)	11.0 (4.0, 26.0)	7.0 (4.0, 10.0)
CXR Severity Score **		3.0 (1.0, 4.0)	3.0 (0.0, 3.0)	3.0 (3.0, 6.0)

BMI: Body Mass Index, COPD: Chronic Obstructive Pulmonary Disease, CHF: Congestive Heart Failure, CKD: Chronic Kidney Disease, DM: Diabetes Mellitus, HIV: Human Immune deficiency Virus, CXR: Chest X-ray; * Mean (Standard deviation); ** Median (25th percentile, 75th percentile).

**Table 2 jcm-11-02157-t002:** Distribution and frequency of radiographic findings.

	*N* (%)
**Radiology Findings**	
No	149 (23.7%)
Yes	479 (76.3%)
**Radiologic Characteristics**	
**Distribution**	
Unilateral Unilobar	64 (13.4%)
Unilateral Multilobar	21 (4.4%)
Bilateral—not diffuse	63 (13.2%)
Diffuse Bilateral	331 (69.1%)
**Opacity**	
Hazy or Interstitial Opacities	357 (74.5%)
Dense Opacities	122 (25.5%)
**Specific Patterns**	
Diffuse-Bilateral with hazy opacities	256 (53.4%)
Diffuse-Bilateral with dense opacities	75 (15.7%)
Unilateral-Unilobar with Hazy opacities	49 (10.2%)
Bilateral with predominance with hazy opacities	37 (7.7%)
Bilateral with predominance with dense opacities	26 (5.4%)
Unilateral-Multilobar with Hazy opacities	15 (3.1%)
Unilateral-Unilobar with dense opacities	15 (3.1%)
Unilateral-Multilobar with dense opacities	6 (1.3%)

**Table 3 jcm-11-02157-t003:** Binary logistic regression analysis of the independent factors predicting mortality in COVID-19 patients.

Variables	Adjusted OR (95% CI)	*p*-Value
Age	1.063 (1.043, 1.083)	<0.001
Gender	1.433 (0.865, 2.374)	0.162
Smoking	0.962 (0.330, 2.799)	0.943
Asthma/COPD	1.045 (0.552, 1.977)	0.893
CHF	0.864 (0.345, 2.165)	0.755
CKD	0.886 (0.426, 1.842)	0.745
DM	1.238 (0.757, 2.023)	0.395
Cancer	1.579 (0.618, 4.037)	0.340
Days from Onset of Symptoms	0.991 (0.936, 1.050)	0.766
Sepsis Syndrome	7.353 (4.434, 12.194)	<0.001
S/F ratio	0.995 (0.993, 0.997)	<0.001
Mechanical Ventilation on Admission	5.389 (2.931, 9.908)	<0.001
CXR Severity Score	1.184 (1.054, 1.330)	0.005

OR: Odds Ratio, CI: Confidence Interval, COPD: Chronic Obstructive Pulmonary Disease, CHF: Congestive Heart Failure, CKD: Chronic Kidney Disease, DM: Diabetes Mellitus, CXR: Chest X-ray.

**Table 4 jcm-11-02157-t004:** Predictive model developed in the derivation dataset (*N* = 439).

Model	Coefficient	*p*-Value	Adjusted OR (95% CI)
Age	0.052	<0.001	1.053 (1.032, 1.074)
Sepsis Syndrome	1.865	<0.001	6.459 (3.667, 11.379)
S/F ratio	−0.005	<0.001	0.995 (0.993, 0.997)
Mechanical Ventilation on Admission	1.606	<0.001	4.981 (2.519, 9.850)
CXR Severity Score	0.132	0.046	1.141 (1.003, 1.299)
Constant	−3.867	<0.001	0.021

OR: Odds Ratio, CI: Confidence Interval, CXR: Chest X-ray.

## Data Availability

The data presented in this study are available on request from the corresponding author.

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
