# Peer review of "Role of a Chest X-ray Severity Score in a Multivariable Predictive Model for Mortality in Patients with COVID-19: A Single-Center, Retrospective Study"

_jcm, 2022, doi:10.3390/jcm11082157_

Round 1
Reviewer 1 Report
Congratulations to the authors for the paper.
The authors made the revisions accordingly. Regarding the suggestions that were not addressed in the revision, I consider that the authors justified their decision with convincing and reasonable justification.
Author Response
Dear respected reviewer,
Thank you very much for your kind words. It was an absolute pleasure working with you, and we very much appreciated your time and all your invaluable suggestions to improve our paper.
Best regards,
Ryan
Reviewer 2 Report
The authors assessed the role of a chest X-ray (CXR) scoring system in a multivariable model for predicting mortality of COVID-19 patients in a single-center retrospective observational study.
The peculiarity of the study is represented by the enrolled population, predominantly comprising of minorities with the highest prevalence of chronic diseases such as hypertension, diabetes, obesity and COPD.
The study is well-organized and presented.
The main limitations are:
-the absence of an ICC calculation: as the score is based on a subjective evaluation of radiological images, the repeatability of the score assignation should be assessed
-the limited references: this topic has been already analyzed in the literature but the references section is poor. I have suggested more references to be added to the manuscript, but others are needed
Abstract
No mention of the number and experience of the readers
Introduction
"Even though Computerized Tomography (CT) scans have the highest sensitivity for characterization of pulmonary involvement in COVID-19 disease, factors such as easy accessibility, higher logistic costs, time management, radiation considerations, and compliance with infection control measures are barriers to its widespread use as the primary imaging modality in the setting of a pandemic" other references about the role of Chest CT in the COVID-19 pandemic should be added (doi: 10.1148/radiol.2020200230.; doi: 10.1007/s11604-020-01010-7; doi: 10.1007/s00330-020-06801-0)
"chest x-ray (CXR) 67 has been shown to play an essential role" add some considerations about the advantages (wide availability, low cost, executions at the patient's bed)
..."some variability in its reported sensitivity" influenced by the timing of the onset of the symptoms
"Multiple retrospective stud-78 ies have evaluated the association between an initial CXR and clinical outcomes in pa-79 tients with COVID-19 pneumonia."Other studies assessed the value of the initial CXR as a prognostic tool, but only two references are available in the text. Valuable references are missing (doi: 10.4269/ajtmh.20-0535., doi: 10.1148/radiol.2020202326)
Results
Please provide a flowchart showing the inclusion and exclusion of the patients, according to the above-cited inclusion and exclusion criteria
"In terms of severity of COVID-19 pneumonia, 41.2% were classified as moderate, 28.3% as severe, and 30.4% as critical." according to which reference? please specify
Author Response
Dear respected reviewer,
Thank you very much for taking the time to review our manuscript and thanks for your invaluable insight and comments. Please below find our point-by-point responses to your questions and comments:
The main limitations are:
-the absence of an ICC calculation: as the score is based on a subjective evaluation of radiological images, the repeatability of the score assignation should be assessed
That is a great suggestion, but unfortunately, as described in the methods section the radiologists provided us with a single consensus score, and we don’t have separate scores available for the three radiologists to be able to run the reliability analysis. The reason for using this method was that we were trying to develop a new scoring system based on chest radiographs, and naturally, it was our radiologists who were going to do the majority of the work in terms of designing the scoring system. We believed that the best way for them to realize the intricacies of the scoring system, figure out where they might have different opinions, and eventually come up with the easiest scoring system with minimal inter-rater variability was to have them discuss all the cases and come up with a consensus. Obviously, research is a dynamic process, and at the time we were thinking that after this introductory phase of the study, we can test the inter-rater variability of our scoring system in the next phases on larger sample populations.
-the limited references: this topic has been already analyzed in the literature but the references section is poor. I have suggested more references to be added to the manuscript, but others are needed
Thank you, the manuscript is revised accordingly.
Abstract
No mention of the number and experience of the readers
Thanks for your recommendation, this is now added to the abstract.
Introduction
"Even though Computerized Tomography (CT) scans have the highest sensitivity for characterization of pulmonary involvement in COVID-19 disease, factors such as easy accessibility, higher logistic costs, time management, radiation considerations, and compliance with infection control measures are barriers to its widespread use as the primary imaging modality in the setting of a pandemic" other references about the role of Chest CT in the COVID-19 pandemic should be added (doi: 10.1148/radiol.2020200230.; doi: 10.1007/s11604-020-01010-7; doi: 10.1007/s00330-020-06801-0)
Thank you very much for recommending these valuable articles, they are now added to our references.
"chest x-ray (CXR) 67 has been shown to play an essential role" add some considerations about the advantages (wide availability, low cost, executions at the patient's bed)
Thanks for the suggestion. These considerations are now added to the manuscript.
..."some variability in its reported sensitivity" influenced by the timing of the onset of the symptoms
Thank you, the sentence is now revised.
"Multiple retrospective stud-78 ies have evaluated the association between an initial CXR and clinical outcomes in pa-79 tients with COVID-19 pneumonia."Other studies assessed the value of the initial CXR as a prognostic tool, but only two references are available in the text. Valuable references are missing (doi: 10.4269/ajtmh.20-0535., doi: 10.4269/ajtmh.20-0535)
Thanks for the recommendation, the manuscript is revised accordingly.
Results
Please provide a flowchart showing the inclusion and exclusion of the patients, according to the above-cited inclusion and exclusion criteria
Thanks for your suggestion, the flowchart is now added to the manuscript.
"In terms of severity of COVID-19 pneumonia, 41.2% were classified as moderate, 28.3% as severe, and 30.4% as critical." according to which reference? please specify
Thank you, it is now specified which criteria was used.
Thanks again for your helpful comments and suggestions. Hope that in light of the new changes, the manuscript is a better fit for publication in your esteemed journal.
Best regards,
Ryan

This manuscript is a resubmission of an earlier submission. The following is a list of the peer review reports and author responses from that submission.
Round 1
Reviewer 1 Report
This is a well written, well structured, very easy to read paper proposing a novel multivariable score to predict mortality in hospitalized patients with COVID-19.However, I have some comments meant to clarify a few aspects (before acceptance):
- The afirmation "This study is the first to have incorporated a CXR severity scoring system in a multivariable model to provide an accurate predictive tool that can be used at bedside on admission" seems to not be true as there was at least one study to provide bedside tools before such as: https://www.sciencedirect.com/science/article/pii/S1201971220303283 that was not cited by the authors (their model combined the predictice factors: Brixia score, patient age, and conditions that induced immunosuppression).
Moreover, there are still more studies including both CXR and other variables to predict COVID-19 outcome in a multivariable model, that the authors did not mention. For instance: https://www.ncbi.nlm.nih.gov/pmc/articles/PMC8550757/
https://www.dovepress.com/artificial-intelligence-in-predicting-clinical-outcome-in-covid-19-pat-peer-reviewed-fulltext-article-RMI
https://www.thelancet.com/journals/landig/article/PIIS2589-7500(21)00039-X/fulltext
Maybe the authors could also integrate in the Discussions the comparisons with these studies and highlight the strengths and weaknesses of the current paper compared to them.
- In Table 1 it would be useful to add another column with the p value in order to hightlight whether there are any significant differences between the "Alive" group and the "Expired group" regarding all the variables presented.
This is important to check for confounding variables.
And also, the authors did not mention how and if they adjusted for confounding variables (for which confounding variables did they do adjustments)
- Although it can be deduced that the primary end-point is the in-hospital mortality, this is not clearly stated nowhere in the text.
Author Response
Dear respected reviewer,
Thank you very much for taking the time to review our manuscript, and thanks for your invaluable insight and comments. Please below find our point-by-point responses to your questions and comments:
- The affirmation "This study is the first to have incorporated a CXR severity scoring system in a multivariable model to provide an accurate predictive tool that can be used at bedside on admission" seems to not be true as there was at least one study to provide bedside tools before such as: https://www.sciencedirect.com/science/article/pii/S1201971220303283 that was not cited by the authors (their model combined the predictice factors: Brixia score, patient age, and conditions that induced immunosuppression).
Thanks for your meticulous attention. You are absolutely correct. The sentence is revised accordingly.
Moreover, there are still more studies including both CXR and other variables to predict COVID-19 outcome in a multivariable model, that the authors did not mention. For instance: https://www.ncbi.nlm.nih.gov/pmc/articles/PMC8550757/
https://www.dovepress.com/artificial-intelligence-in-predicting-clinical-outcome-in-covid-19-pat-peer-reviewed-fulltext-article-RMI
https://www.thelancet.com/journals/landig/article/PIIS2589-7500(21)00039-X/fulltext
Maybe the authors could also integrate in the Discussions the comparisons with these studies and highlight the strengths and weaknesses of the current paper compared to them.
Thank you very much for pointing these studies out. The two with comparable methods and outcomes are now added to our discussion section. The significant difference in methods, and also the outcome of Esposito’s study makes it difficult to be directly compared to our results. Jiao’s paper also had a similar issue where they used deep learning models to analyze the images, so their findings could not be directly compared to ours, therefore we did not include these two papers in our comparisons. Please let us know if you feel like they should definitely be included in our discussion.
- In Table 1 it would be useful to add another column with the p value in order to highlight whether there are any significant differences between the "Alive" group and the "Expired group" regarding all the variables presented.
This is important to check for confounding variables.
And also, the authors did not mention how and if they adjusted for confounding variables (for which confounding variables did they do adjustments)
Thank you for your great suggestion. We agree that this information will be interesting to look at, and this was something that we were also interested in doing. But after a long discussion with our statistician, he convinced us that these comparisons are not the main objective of the study and performing them only subjects our analyses to “multiple comparisons” which will then need to be addressed by adjusting the p value using for example the Bonferroni correction, or other statistical methods, which is not an optimal analytic strategy. We argued that most studies similar to ours have presented p values for all their univariate comparisons, without any modification of their statistical methods. But he insisted that from a statistical standpoint, it is not academically correct, and suggested we do not perform univariate comparisons. As for your second comment, basically, the confounding factors we thought might be valid in these analyses (considering the number of variables we could have included in our regression analysis based on our sample size) were the variables included in the regression models (presented in table 3). And the adjusted ORs in that table represents the contribution of each of those variables controlled for the rest of the variables included in the model. Our statistician strongly recommended against performing univariate analyses, and choosing our possible confounding factors based on the p values of those univariate analyses. He explained that the best method would be selecting the confounding factors based on our clinical knowledge and available literature, which was what we did for this study. Please let us know if you think performing those univariate comparisons and adding the p values are necessary. We made some revisions to clarify the confounding factors.
- Although it can be deduced that the primary end-point is the in-hospital mortality, this is not clearly stated nowhere in the text.\
Thanks for pointing this out. Revisions were made to address this.
Thanks again for your helpful comments and suggestions. Please let us know what you think about the suggestions that were not addressed in this revision. We are more than happy to make additional changes to the manuscript. Hope that in light of the new revisions, the manuscript is better fit for publication in your esteemed journal.
Best regards,
Ryan

Reviewer 2 Report
- Although the paper is well-written and methodologically sound, I have some rather fundamental concerns. Although there is indeed a lot of potential value to be found in a predictive model for COVID patients that does not rely on CT scans, the patients in this study were recruited between Mar and Apr 2020, i.e. several COVID-19 variants ago. The predictive value of the model on the Omicron variant (which currently constitutes 100% of the cases in the authors’ country) is highly questionable. Furthermore, a large body of work has been published regarding the use of chest X-rays in COVID assessment; it is not clear what added value this study provides considering the timeline of patient recruitment as well as the fact that it is a single-center study.
- I am not sure if a model predicting mortality can be directly compared to one predicting severity.
- Although the authors mention it as a limitation, the fact that you have three radiologists evaluating images in consensus does not reflect the clinical reality. It has been shown that models that rely on subjective radiological interpretation are quite sensitive to inter- and intra-observer agreement.
- I was somewhat surprised to see 15 authors for a single-center study involving conventional ML modeling. Did all authors provide an essential contribution?
- A comparison between different types of ML models (decision tree/RF, NN, etc) could have been interesting
Author Response
Dear respected reviewer,
Thank you very much for taking the time to review our manuscript, and thanks for your invaluable insight and comments. Please below find our point-by-point responses to your questions and comments:
Although the paper is well-written and methodologically sound, I have some rather fundamental concerns. Although there is indeed a lot of potential value to be found in a predictive model for COVID patients that does not rely on CT scans, the patients in this study were recruited between Mar and Apr 2020, i.e. several COVID-19 variants ago. The predictive value of the model on the Omicron variant (which currently constitutes 100% of the cases in the authors’ country) is highly questionable. Furthermore, a large body of work has been published regarding the use of chest X-rays in COVID assessment; it is not clear what added value this study provides considering the timeline of patient recruitment as well as the fact that it is a single-center study.
Thank you for raising all these valid concerns. Unfortunately, this is an inherent issue with performing research studies. At the time of designing and implementing this project, we had no idea that multiple new variants of the virus will emerge. Considering all the time and energy put into finishing this study we did not want to shelve this work. Having said that, we also believe that although the behavior of new COVID variants have been quite different from the initial wave; however, the variables we used for developing our model seem to be generalizable to the new variants too, since they are mostly clinical indicators of a poor outcome in patients with acute respiratory diseases. For instance, it is reasonable to think that increased age, increased oxygen requirement, need for mechanical ventilation and sepsis syndrome would increase the risk of mortality in any respiratory illness, or even non respiratory acute conditions. We hope to be able to test our model on larger patient populations afflicted with new COVID variants in the near future, but unfortunately at this time, we don’t have the data readily available to be able to run the analysis and present the results in the current manuscript. Regarding your concern about what our model is adding to the currently published body of work, we believe our model has had a very high predictive performance for mortality of the patients, even compared to other available models, as mentioned in our discussion section. We also used readily available clinical data, rather than various lab values or deep learning-based image analyses. You are absolutely on-point with the limitation of the data coming from a single center. Hopefully we will be able to test the model on multicenter data soon. We have made some revisions in the manuscript to point out some of the concerns you raised here.
I am not sure if a model predicting mortality can be directly compared to one predicting severity.
Thanks for raising this concern. Sounds reasonable, but generally speaking, one of the main reasons for trying to define different levels of severity for a condition is likely to determine which patients would do worse and will benefit from more aggressive interventions. So, in this study, we did not compare our model to the COVID severity score as a known predictor of mortality, rather we wanted to compare our model to a more widely used severity score for COVID, which has shown to be a good predictor of mortality as well. This severity score had an AUC of 0.85 in our small sample of 185 patients, which is quite remarkable even compared to predictive values reported for published models developed specifically for COVID patient mortality. We are more than happy to remove that comparison from the paper, but since we did not have all the variables needed to compute any of the other published models for mortality prediction, we decided to compare our model to this severity score and we believe this gives the readers a better perspective of our findings, rather than just looking at the numbers without any comparison. We made some revisions to acknowledge that this variable was not specifically designed to predict mortality.
Although the authors mention it as a limitation, the fact that you have three radiologists evaluating images in consensus does not reflect the clinical reality. It has been shown that models that rely on subjective radiological interpretation are quite sensitive to inter- and intra-observer agreement.
Again, your concern is completely valid and we agree with you. The reason for using this method was that we were trying to develop a new scoring system based on chest radiographs, and naturally it was our radiologists who were going to do the majority of the work in terms of designing the scoring system. We believed that the best way for them to realize the intricacies of the scoring system, figure out where they might have different opinions, and eventually come up with the easiest scoring system with minimal inter-rater variability was to have them discuss all the cases and come up with a consensus. Obviously, research is a dynamic process, and at the time we were thinking that after this introductory phase of the study, we can test the inter-rater variability of our scoring system in the next phases on larger sample populations.
I was somewhat surprised to see 15 authors for a single-center study involving conventional ML modeling. Did all authors provide an essential contribution?
That is a fair observation. We assure you that the list of authors has been reviewed multiple times to make sure everyone has made major contributions to the study. The main challenge with this project was basically the timing, which was during a crisis, and having our colleagues add to their workload for helping us with data collection and other aspects of this project was a huge ask. So, because of the general time-constraints of the situation, each person might have allocated less time when compared to authors of a similar study not performed during a pandemic, but their contribution has been instrumental in conducting this project, and removing their names would not be fair.
A comparison between different types of ML models (decision tree/RF, NN, etc) could have been interesting
You are absolutely right, that would be a very interesting comparison, but it seems to be quite outside the scope of the current study, which is an introduction to our new model. Hopefully in the near future, we might be able to perform a more comprehensive study on our model, using a more inclusive sample population with newer COVID variants and comparing it to the best non ML and ML models developed.
Thanks again for your helpful comments and suggestions. Please let us know what you think about the suggestions that were not addressed in this revision. We are more than happy to make additional changes to the manuscript. Hope that in light of the new changes, the manuscript is better fit for publication in your esteemed journal.
Best regards,
Ryan
